# Machine Learning for Patient-Based Real-Time Quality Control (PBRTQC), Analytical and Preanalytical Error Detection in Clinical Laboratory

**DOI:** 10.3390/diagnostics14161808

**Published:** 2024-08-20

**Authors:** Nathan Lorde, Shivani Mahapatra, Tejas Kalaria

**Affiliations:** Blood Sciences, Black Country Pathology Services, The Royal Wolverhampton NHS Trust, Wolverhampton WV10 0QP, UK

**Keywords:** machine learning, artificial intelligence, PBRTQC, quality control, laboratory error, bias

## Abstract

The rapidly evolving field of machine learning (ML), along with artificial intelligence in a broad sense, is revolutionising many areas of healthcare, including laboratory medicine. The amalgamation of the fields of ML and patient-based real-time quality control (PBRTQC) processes could improve the traditional PBRTQC and error detection algorithms in the laboratory. This narrative review discusses published studies on using ML for the detection of systematic errors, non-systematic errors, and combinations of different types of errors in clinical laboratories. The studies discussed used ML for detecting bias, the requirement for re-calibration, samples contaminated with intravenous fluid or EDTA, delayed sample analysis, wrong-blood-in-tube errors, interference or a combination of different types of errors, by comparing the performance of ML models with human validators or traditional PBRTQC algorithms. Advantages, limitations, the creation of standardised ML models, ethical and regulatory aspects and potential future developments have also been discussed in brief.

## 1. Introduction

Clinical laboratories face ever-increasing test demand, with an annual 8% to 10% increase in the workload [1]. In addition, some healthcare systems have opted for consolidated pathology service models to improve efficiency [2]. These changes in demand and operating model lead to large laboratories housing numerous analysers for the same set of tests, operating in parallel. One of the challenges of such transformation, compared to smaller laboratories, is the risk of human technical and clinical validators becoming overwhelmed by the volume and turnover of the data. Though the human brain can hold vast amounts of information, limitations in the amount of information it can process and the speed at which information is processed are well known. With the increasing complexity of input (for example, more parallel data streams, wider test repertoire, wider patient pool), information processing time lengthens (Hick’s Law) [3]. The manual result authorisation, as traditionally employed, of large volumes of such complex laboratory data is challenging and could lead to decreased chances of error detection and increased risk of releasing erroneous results, or hinder progress towards improvements in error detection (Figure 1).

All clinical laboratories have internal quality control (IQC) programmes using the measurement of quality control (QC) samples at specified times. However, this approach has several limitations. To list a few: IQC is retrospective rather than real-time, and therefore, a significant change may be detected several hours later. The assumption inherent to the IQC programme that errors are sustained until the next QC run is known not to be the case for all errors. In addition, the cost of obtaining QC material for each test and the occasional lack of commutability of QC materials pose additional challenges [4,5,6]. The retrospective nature of traditional IQC particularly can result in the need to repeat many or all analyses up to the point of the last satisfactory IQC result, leading to increased burden on laboratory staff, including the possibility of having to directly contact clinicians if a repeat analysis gives a significantly different result from the one released before. In addition, traditional forms of IQC do not assess other aspects of the total testing process, such as sample transportation issues in the preanalytical phase.

Thoughtfully harnessed patient data could contribute to quality control and pattern recognition. Patient-based real-time quality control (PBRTQC) is a technique that uses the long-term monitoring of patient data through calculations such as moving average (MA), moving standard deviation (MovSD) or moving median (MM) for each analyte. PBRTQC is often superior to periodic IQC in the detection of newly introduced bias or imprecision, and therefore it is particularly useful for analytes that are difficult to keep in control for prolonged periods of time [4,7,8]. PBRTQC requires more complex modelling than simple IQC charting, as changes in requesting practices over the course of a day or week may impact any real-time metric. PBRTQC is not suitable for every test, such as low throughput, qualitative or semi-quantitative results, and IQC runs will remain important in scenarios such as post-maintenance or post-calibration checks [4,8]. In addition, the simple monitoring of summary statistics, such as MA or MM must allow for sufficiently wide limits, so that the noise from random variation does not lead to unnecessary stoppages and investigations. However, these limits must still be tight enough to detect significant systematic errors [4,8,9]. Another key limitation is that random point errors are usually not detected by most current PBTRTQC methods [10].

There has been a big leap forward in the integration of machine learning (ML), a subfield of artificial intelligence (AI), in a variety of industries, including the healthcare sector. ML encompasses a range of algorithms that allow computer systems to perform processes to sort data or use data for prediction in ways not directly programmed by humans [11,12,13,14]. The computer programme will “learn” to handle and analyse data and then create an output. There are a vast number of different ML models that can broadly be divided into supervised and unsupervised learning techniques [12,13,15]. Supervised learning models are trained using input data with labelled outputs and allowed to “learn” how to sort the inputs into these pre-defined outputs. The labelled output data are said to “supervise” learning [11,12,13,15]. Supervised learning algorithms include regression analyses, decision trees (DTs), random forest (RF), support vector machine (SVM), gradient boosting methods, k-nearest neighbours (k-NN), Bayesian networks (BN) and neural networks (NN) [15]. Unsupervised ML models are those that use unlabelled data and sort data into inherent groups using patterns that a human may not be able to see [15]. Examples include principal components analysis (PCA) and k-means clustering [15]. There are also semi-supervised learning models, which lie between supervised and unsupervised learning models. ML models are ever growing and the above list is by no means exhaustive [15,16].

ML models have been used in clinical laboratories for prediction of sample turnaround times [17], creation of reference intervals [18], predicting ferritin concentrations [19,20], predicting test number and machine workload [21], managing reagent stocks [21], predicting a diagnosis of COVID-19 [22,23,24,25,26] and for predicting patients’ most likely diagnoses [27]. ML has also been trialled by many laboratories for PBRTQC and for error detection in recent years. For ease of understanding, in this narrative review we have classified studies employing ML in the field into those identifying systematic, non-systematic or a combination of different types of errors.

## 2. ML for Detection of Systematic Errors

A relatively early example from 2016 is by Sampson et al. using a logistic regression algorithm to create error scores and then monitoring the cumulative sum of the error score [28]. It was called CUSUM Logistic Regression (CSLR). In this model 14 chemistry analytes were measured, namely sodium, potassium, chloride, urea, creatinine, bicarbonate, alkaline phosphatase (ALP), alanine transaminase (ALT), aspartate transaminase (AST), glucose, albumin, calcium, total protein and total bilirubin, with log-transformation if not normally distributed. Then the result of each analyte was predicted using a stepwise linear multiple regression that incorporated the results of the other 13 analytes in the sample as the independent variables, using 179,280 results collected over three years. The predicted and actual results for each analyte, day of the week and the time of the day were combined in a logistic regression model to generate a probability of error in the assay for that analyte. The logistic regression model was trained using 53,607 results captured from one year. They randomly transformed half of these to simulate biases. The probabilities, minus the mean probability score of “good” data, were added cumulatively over the day (score started at 0 each midnight) and when the score reached a specified threshold, different for each analyte, an investigation of significant bias was triggered. The thresholds were set at levels to create a specificity of 90% for detection of errors. The inclusion of time of day and day of week in the regression model was to allow for the significant changes in request trends in some analytes. The full CSLR had a lower number of samples tested once a systematic error was intentionally incorporated into the data until detection of the error, compared with the simpler regression which included only the analyte results as independent variables. The full model detected errors in 7 to 80 samples and found 98% of all simulated errors in albumin as an example, compared with 87 to 172 sample run length for the simpler model, which could only detect 61% of the albumin biases. The authors proposed either using the CSLR to alert to the possibility of a bias that can then be investigated with standard IQC testing or of using it to help decide when to run QC material, instead of having to perform IQC at fixed times.

Duan et al., in a study published in 2021, collected patient data for four chemistry analytes: sodium, chloride, ALT and creatinine, initially passed through winsorization, transformed by Box-Cox and then calculated three different PBRTQC algorithms: moving average (MA), exponentially weighted moving average (EWMA) and moving standard deviation (MovSD) [29]. The authors then created what they called Regression-Adjusted Real-Time Quality Control (RARTQC). For this, the respective PBRTQC model had incorporated into it a multiple regression that included patient sex, outpatient versus inpatient status, requesting department information and diagnosis information. Thus, for each analyte, there were three PBRTQC models and three RARTQC models. All six models were tested for the detection of constant and proportional bias, as well as random error. For random error, the simulated error was added to individual results, while for the biases, the entire data streams were altered to simulate an assay drift. Over the course of a year, 79,587 sodium results, 79,588 chloride results, 328,883 ALT results and 418,494 results for creatinine were obtained, with the data from the first nine months being used for training, while the data from the last three months were used for testing. The RARTQC model based on EWMA was the quickest to identify intentionally introduced systematic bias. The authors identified that no model was good at detecting random error. The models were compared using the “trimmed average of number of patients affected before detection” (tNAPed) at different bias levels and plotted against the bias introduced. The total allowable error was 2% for sodium, 5% for chloride, 16% for ALT and 12% for creatinine. The best constant error average tNAPed values were 56.5 by RARTQC EWMA for sodium, 7.5 for chloride by RARTQC MA, 51.5 for ALT by RARTQC EWMA and 56.2 for creatinine by RARTQC EWMA at the total allowable error for each analyte. The tNAPed for the best performing non-RARTQC models were 83.4 by EWMA for sodium, 11.0 by EWMA for chloride, 126.0 for ALT by EWMA and 199.8 for creatinine by EWMA. The RARTQC models similarly outperformed the non-RARTQC models in proportional bias detection. The study demonstrated that ML PBRTQC algorithms require the data to be optimised first in order to achieve good results. The authors trialled different block sizes, which are the numbers of preceding results that are included in the calculation of the moving parameter at any one time, and the truncation limits, which are the limits outside of which the outlier results are excluded. These are called hyperparameters. Most ML and PBRTQC algorithms will have some hyperparameters that require “tuning” to optimise performance.

Zhou et al. developed what they called machine learning internal quality control (MLiQC) [30]. The authors used data from five chemistry analytes and obtained 1,195,000 data points from a Siemens ADVIA^®^ 2400. The analytes were albumin, AST, ALT, glucose and total protein. Of the total dataset, 956,000 data points were used in training and 239,000 were used to test the model for the detection of intentionally introduced biases. The random forest (RF) model was able to detect systematic biases in fewer samples than the four PBRTQC models tested—EWMA, MA, moving median (MM) and the Harrell–Davis median (HD50). The number of results with bias missed before the bias was detected and the accuracy of the MLiQC and PBRTQC models at different bias thresholds were assessed, including the “critical bias” (elsewhere known as the desirable bias limit [31]), calculated as 0.25 × √((CVi)2 + (CVg)2), where CVi is the intra-individual coefficient of variation and CVg is the between-individual coefficient of variation. The critical biases were 1.3% for albumin, 5.4% for AST, 12% for ALT, 2.5% for glucose and 1.2% for total protein. MLiQC achieved an area under the receiver operating characteristics curve (AUROC) at these biases of 0.9848, 0.9927, 0.9889, 0.9946 and 0.9816, respectively. The average number of patient samples from the point a bias was introduced until it was detected were between 8 and 22 using MLiQC at critical errors, whereas they were from 50 to >1000 for traditional PBRTQC.

Zhou et al. also developed other ML algorithms to augment the PBRTQC procedures. An ML algorithm was created combining RF, a support vector machine (SVM) and an NN to detect deliberately introduced biases in prostate specific antigen (PSA) results [32]. PSA results from 5½ years from one centre were collected, excluding the outlying 1%. The training dataset comprised 43,699 results and the testing dataset 39,700 results. The 43,699 results in the training dataset were augmented by randomly adding a number in the range of −0.3 to +0.3 µg/L to each result, doubling the pool of results. This process was repeated for a total of 80 rounds creating a dataset of 3,495,920 results, which served as non-biased data. To each of these 3,495,920 datapoints, either 0.02 µg/L or 0.15 µg/L was added to create two different streams of biased data for training. Using the combined error-free and biased training data points, each of the ML models (SVM, RF and NN) were trained, and five-fold cross-validation was performed to determine the average accuracy of each. The error from the prediction of each individual model was used to determine the individual model’s proportion of the sum of the error of all three. This proportion of the error of each model was used to calculate the weight given to the model’s score to calculate the overall probability score for the fusion of the models. SVM, RF and NN were weighted 0.34, 0.27 and 0.39, respectively. In the test dataset, they added biases of 0.01, 0.03, 0.05, 0.08, 0.10 and 0.20 µg/L. This fusion ML model performed very well against the best performing PBRTQC algorithm, which, in this case, was a moving sum of the outputs (MovSO), at all levels of bias. Taking −0.1 and +0.1 µg/L biases, the optimal MovSO algorithm had a median number of patients missed of 157 and 245, versus 9.5 for both biases by the fusion ML model.

In another study by the group, Liang et al. employed an RF model, called machine learning quality control (MLQC), to detect performance shift in seven full blood count parameters, namely lymphocyte count, lymphocyte ratio, haemoglobin, mean corpuscular haemoglobin, mean corpuscular haemoglobin concentration, red blood cell distribution width and platelet count [33]. Total 423,290 results were obtained over one year, with the first ten months used for training and the last two months for validation. A further 22,460 results from another centre were used for testing. The results below the 25th centile and above the 75th centile were removed, and so were the results from patients with only one sample during the study period, after which intentional biases were introduced. The delta data for the individual patients were recorded. The Isolation Forest (IF) algorithm was used to pre-process the data before passing them to the RF model, an action the authors showed improved the discriminating ability of the MLQC model. MLQC, using delta data, was able to detect biases faster and had superior accuracy compared to the two PBRTQC models (MA and MovSD), which analysed both the numerical results and the delta data. Lymphocyte count, for example, had a median number of patients missed of five before the error was detected by MLQC at the critical bias, versus 72 for the best performing PBRTQC, which was MovSD using delta data.

Liang et al., in another study by the previously described group, used a classification and regression tree (CART) model to detect intentionally introduced biases in 10 measurands, including white cell count, red cell count, haematocrit, haemoglobin, platelet count, AST, ALT, glucose, total protein and albumin [34]. The results from 11 hospitals were collected, with eight used for training and three for testing. In addition to the test results, laboratory identification, department, out- versus in-patient, diagnosis, sex, age, the date of the sample, the time of the sample, the brand of the analyser, the unique identification of the analyser and the units the result was issued in were collected. The training dataset contained 3,097,661 chemistry datapoints and 6,786,110 haematology points. Fifty biases were introduced in 50 steps, from −50% to +50%. This model was called machine-learning nonlinear regression-adjusted patient-based real-time quality-control (mNL-PBRTQC). The authors compared the model’s performance against a basic PBRTQC (EWMA), as well as a linear regression-adjusted real-time quality control (L-RARTQC). The mNL-PBRTQC mostly outperformed the other two in terms of number of samples missed and accurate delineation of biased samples. The CART’s outputs served as the input of an EWMA in this model. The best performing L-RARTQC used four non-result variables and the best mNL-PBRTQC used eight. The red blood cell count was the only result where mNL-PBRTQC performed worse compared to the other models. For this parameter, the median number of samples missed was 2460 for mNL-PBRTQC versus 2298 for the plain EWMA and 2415 for L-RARTQC across the range of biases. The median number of patients missed was lower for the mNL-PBRTQC for all other analytes, though at some specific bias levels in some analytes, it had a slightly higher number of missed patient samples compared to the L-RARTQC model.

Regulski et al. recently described different applications for ML in their network of medical laboratories [21]. One of the applications tested involved using a multilayer perceptron (MLP), a class of NN, to detect significant assay drift and therefore the need for re-calibration. The best performing MLP in this study was configured with nine input neurons, one hidden layer with 13 neurons and an output layer with two neurons. It was developed using 16,500 measurement results and 13 calibration points. The MLP performed better than a DT model that was also tested. The nine input neurons used in the MLP were MA, natural logarithms of moving median (ln(MME)), square-root of MA (Sqrt (MA)), average of normals (AoN), natural logarithm of MA (ln(MA)), square root of MME (sqrt (MME)), natural log of the result (ln(x).r) and square-root of the result (Sqrt(x).r). The best performing MLP missed only 1.5% of out-of-control assay drifts that required re-calibration. The authors discussed that this ML model would allow less frequent re-calibrations and thus reduce wastage compared with re-calibrations after a failed internal quality control result. They showed that IQC tests could fail despite there being no error in the patient results being output at the same time. Thus, reliance on IQC alone would result in re-calibration events when they are not needed. This MLP was part of a prototype suite of ML models to optimise laboratory performance, including predicting future request patterns, predicting the need for ordering reagents and predicting the load on various analysers within the laboratories.

In all the above examples of using ML to augment PBRTQC for systematic error detection, except in the studies by Sampson et al. [28] and Regulski et al. [21], the ML augmented algorithms were compared with standard PBRTQC dataflows. In each of these cases, the ML-augmented algorithms were able to detect systematic errors faster than traditional PBRTQC, which, if these results are generalizable, would result in fewer patients’ samples being affected by an assay drift before the drift is detected.

## 3. ML for Detection of Non-Systematic Errors

### 3.1. Intravenous Fluid Contamination

Baron et al. used DTs to create an algorithm to detect spuriously high glucose from contamination with glucose-containing intravenous fluid [35]. They retrospectively collected 6 months of reported glucose concentrations of >28 mmol/L, as well as the corresponding records for glucose results before and after the high result, the presence of a diagnosis of diabetes mellitus and any record of intravenous fluid or parenteral nutrition around the time of the high result. Expert review of all these factors was used to label samples as contaminated versus not contaminated. One DT was trained using the coexisting analytes collected from the same sample (sodium, potassium, chloride, bicarbonate and anion gap) as well as the patients’ glucose results in the preceding year as inputs. This DT revealed that the most significant factors in determining whether a high glucose level was due to intravenous fluid contamination was a glucose result >44 mmol/L and an anion gap ≤15 mmol/L. This information was used to create an algorithm for the laboratory staff. Though the DT was not promoted as something to be used “live”, the authors stated this efficient data mining for feature selection was greatly beneficial. Upon implementation, the algorithm identified only 14 out of 43 samples with spurious high glucose because of intravenous fluid contamination, when concentrations >28 mmol/L were evaluated (33% sensitivity, 100% specificity). However, this rose to 14 out of 19 samples with spurious high glucose >44 mmol/L (74% sensitivity).

Spies et al. used an unsupervised approach called Uniform Manifold Approximation and Projection (UMAP) for intravenous fluid contamination detection [36]. The results of sodium, potassium, chloride, blood urea nitrogen, total CO_2_, creatinine, glucose, calcium and calculated anion gap for 2,567,403 samples from 312,721 inpatients were collected. Simulated contaminated samples were created by theoretically adding various proportions of 10 commonly used intravenous fluids to 54,000 patient samples. A training dataset was created by adding these 54,000 simulated contaminated samples to 1,620,275 real-life samples. The remaining collected data were used as the testing dataset. If the UMAP prediction agreed with the interpretation given by the laboratory staff, then it was deemed accurate. For cases of non-concordance, a laboratory expert reviewed the results and the patient information to adjudicate. The area under the precision recall curve was 0.89 for the UMAP prediction. Unlike Baron et al. [35], Spies et al. proposed the real-time deployment of the model.

### 3.2. Wrong-Blood-in-Tube (WBIT) Errors

Strylewicz and Doctor used data from the Action to Control Cardiovascular Risk in Diabetes (ACCORD) trial, in which separate samples for glucose and HbA1c were collected from participants [37]. WBIT errors were introduced in 50% of the 8-month samples in the dataset, changing some HbA1c results by +/−1.0% and some glucose results by +/−3.89 mmol/L. A probabilistic error detection using BN was built to detect the WBIT events. The BN outperformed the laboratory experts in identifying the simulated errors, with an AUROC of 0.79 compared to an AUROC of up to 0.73 for the experts.

Doctor and Strylewicz used another BN to detect intentional errors in HbA1c and glucose results in data from the National Health and Nutrition Examination Survey (NHANES) and the Diabetes Prevention Program (DPP) studies [38]. The ability of the BN to detect mismatches was compared against a commercially available automated algorithm in the NHANES data and against 11 laboratory experts in the DPP data. The BN outperformed the commercial algorithm and 7 out of 11 experts, and in no case was it inferior.

Rosenbaum and Baron used ML multianalyte delta checks to identify WBIT [39]. The results of 11 analytes for patients, who had to have all 11 analytes tested twice in 36 h, were collected. Some of the more recent samples were switched among patients to simulate WBIT. A logistic regression model and a SVM that had as its inputs the absolute difference between the results, the actual values of the results, the velocity of change of the results and combinations of these were trained and tested. The SVM and logistic regression both had greater AUROC for error detection compared to a method employing only the delta checks of individual analytes. They then deployed the SVM model on biochemistry laboratory results in real time. Of the 217 samples flagged by the SVM, the authors reviewed the electronic health records of 23 patients and concluded that the majority were explainable by patient factors, such as dialysis. Even though this would suggest poor positive predictive value, the authors argued that use of their ML model will flag only those samples at high risk of WBIT, which would be fewer than those flagged by their standard delta checks and would reduce the number of samples that a laboratory medical professional would have to review.

Farrell tested eight different ML models against human performance for the detection of WBIT [40]. Patients with at least two sets of results of six biochemistry analytes in seven days were included, and half had their most recent samples exchanged to simulate WBIT. All eight of these models—an artificial neural network (ANN), extreme gradient boosting (XGB), SVM, RF, logistic regression, k-NN, simple DT and complex DT—outperformed the 50 laboratory professionals in accurately detecting the errors. The ANN performed the best, with an accuracy of 92.1 ± 0.44% and the simple DT, which was the lowest performing ML model, had an accuracy of 86.5 ± 0.25%. The laboratory professionals had an accuracy of 77.8 ± 4.6%.

Zhou et al. tested six ML delta check classifiers with different algorithms for WBIT detection [41]. A deep belief network (DBN), RF, SVM, logistic regression, k-NN and a Naïve Bayesian Classifier were assessed in this study. Deidentified haematology results were divided into training and validation datasets, and the results from another laboratory were used as a test dataset. WBIT was simulated by switching the results of haematology tests. The models were compared with one another and with an empirical delta check, a revised weighted delta check and the use of reference change value (RCV)-based delta check. The data input as a group of 22 haematology tests led to a more accurate WBIT error detection compared to the input of individual test results. The DBN, using absolute delta values for 22 of the full blood count parameters, performed the best in the test dataset (accuracy 93.1%, AUROC 0.977).

Mitani et al. collected sample data in their laboratory, in which 11 to 15 of specified biochemistry and haematology analytes were measured, and for which the patient had at least three preceding samples with 11 to 15 of these specified analytes measured as well [42]. Random shuffling of the latest results was carried out to simulate WBIT. A gradient-boosted-decision-tree (GBDT) was trained and then compared with use of delta checks and with an index calculated from the delta data for each analyte called the weighted calculated difference index (wCDI). In the testing dataset, the AUROCs were 0.9984, 0.9378 and 0.9818 for the GBDT ML model, the delta checks and wCDI, respectively. The positive predictive values were 0.4353, 0.0037 and 0.0196, respectively.

### 3.3. Interference

In a study to assess serum quality, Yang et al. assessed a deep learning convolutional NN model trained on a dataset of centrifuged blood sample images with known serum haemolysis, icteric and lipaemic indices [43]. The image recognition ML model achieved AUROCs of 0.989, 0.996 and 0.993 for sample subclassification, with the analyser-measured haemolysis, icteric and lipaemic indices, respectively, being used as the underlying indicators. The argument for this use case was that the image recognition would subclassify the samples faster than the measurement of indices by the automated analyser, with only negligible reduction in accuracy.

## 4. ML for Detection of Combinations of Errors Simultaneously

Demirci et al. used an ANN to determine whether samples could be autoverified without human intervention in 2016 [44]. The consensus of seven laboratory specialists was obtained to create rules to verify 13 chemistry analytes: sodium, potassium, chloride, calcium, magnesium, glucose, urea, creatinine, AST, ALT, gamma-glutamyl transferase (GGT), ALP and uric acid. The input parameters were the results of these analytes, as well as delta sodium, delta potassium, delta calcium, delta magnesium, delta glucose, delta uric acid, delta chloride, patient age, the haemolysis index, the lipaemic index and the icteric index. The errors that were being detected were the following: interference by a serum index above threshold for the respective analyte, significant delta change that has no clinical explanation, intravenous fluid contamination, EDTA contamination, delayed sample analysis causing spurious results and results incompatible with life. They first created reference rules for accepting or rejecting results and then selected 1847 patient samples with at least 50% of the 24 inputs required from historically analysed samples to test against these rules. At the optimum input parameters, the ANN was compared initially with the real-time evaluation of 3829 samples by laboratory specialists. The ANN had a sensitivity of only 27.8%, but a specificity of 99.7% for the detection of invalid results. However, when the samples with discordance between the ANN and the human validator were re-analysed by laboratory specialists without time pressure, they found that 127 results that were accepted as valid by the human in real time but marked invalid by the ANN were indeed invalid, and, similarly, nine samples rejected by the human but marked as valid by the ANN were in fact valid. When these alterations were made, the sensitivity of the ANN for detecting errors causing invalid results was 91%, and the specificity was 100%. A DT was also created in this study. However, the DT did not perform as well, with a sensitivity of 48.9% and a specificity of 99.9% on the same test dataset.

Wang et al. created an ensemble ML model for autoverification [45]. Four base models were trained and tested: a Naïve Bayes Classifier, k-NN, RF and XGBoost; and 52 biochemistry analytes were included. Given the large number of analytes, there was a high number of missing results. They used three different methods of handling this “missingness” with each of the four models: replacing the missing result with the population median of that analyte, replacing it with the mean of the reference interval and thirdly, performing PCA to condense the data prior to introducing them to the ML algorithms. PCA is an unsupervised ML model that summarises the variation in a set of observations to one or more principal components. A principal component will not be a particular analyte, but a kind of summary of the effect of combinations of analytes. This process in effect created 12 different models of ML combined with “missingness” correction. They trained and tested the models initially using a dataset of 36,500 real life samples and then performed external testing on a further dataset of 13,355 samples. The pre-labelling was performed by three laboratory medical practitioners by consensus. Invalid results were either those that needed review of the patient’s clinical scenario and possibly discussion with the clinician looking after the patient or those which were implausible and would require checking for analytical anomalies. In this first round of testing, the authors ranked the methods by their AUROCs and showed that the models based on XGBoost performed the best (AUC 0.972 to 0.98). However, the false negative rates were high (44.1% to 68.2%). The authors believed that the initial training set was too imbalanced, the valid samples vastly outnumbered the invalid ones, and therefore, the models were retrained using more data (61,617 samples) as well as by performing oversampling. Two methods were used for each ML model and “missingness” strategy combination. These were the Synthetic Minority Over-sampling Technique (SMOTE) and the Adaptive Synthetic Sampling Approach (ADASYN). SMOTE works by creating synthetic invalid results by choosing an invalid sample at random and then selecting its k-nearest neighbours and creating the synthetic cases so that their results lie between the randomly sampled point and its neighbours. ADASYN similarly creates synthetic invalid results but instead of selecting the invalid samples at random, it chooses them based on the ones that are the most represented in the invalid class, calculating a density function of the invalid class first in order to do so. The models trained with these modifications were tested on 21,063 samples, and this time, the models with the lowest false negative rates were ranked highest. The three top models this time around all had a false negative rate of 2.743%. These models were the following: XGBoost with the population median used to replace missing values and ADASYN as the oversampling method (AUC 0.982); XGBoost with the reference range mean as the missing imputation value and SMOTE for oversampling (AUC 0.981); and thirdly the RF model with the population median for missing data and ADASYN for oversampling (AUC 0.953). The final ensemble ML averaged the output scores of these three models. The samples with an average score > 50 and ≤50 were judged as invalid and valid, respectively. This final ensemble ML model had a false negative rate of 0.411% and an AUROC of 0.998. This ensemble ML model was assessed on real-life results by comparing the model’s efficiency with the laboratory’s current standard practice of using a rules-based autoverification algorithm. The ensemble ML marked fewer samples as potentially invalid. The rules-based system passed 50.2 to 65.1% of samples as valid in each 30 min window, while the ML programme passed 87 to 94% of samples. The authors stated that this ML ensemble would greatly improve laboratory efficiency, as fewer samples would need to be evaluated by a human than is currently the case with the rules-based algorithm. The low false negative rate of 0.411% indicated that this greater efficiency only came with a small risk of missing invalid samples.

## 5. Standardised ML Model Creation and Ethical Considerations

Many studies have emphasised the importance of data optimisation, including block size, as a preliminary step [29,30,33,34,41]. Significant work is required to identify and optimise the hyperparameters for the ML algorithms. An inherent risk is that of overfitting; that is, tuning the hyperparameters to create optimum performance in one dataset could make the ML model perform less optimally in another dataset [15]. ML models require substantial amounts of data in a suitable format to train. It is important that the data used are representative of the population/environment the model is to be used in. A large, diverse and representative training dataset is of utmost importance, and the use of best practice recommendations and checklists for the ethical design of the ML systems could help to maximise the fairness, equity and robustness of the models [46,47].

The selection of the model type, the construction of the data set, data sampling and data preprocessing are some of the factors that could have significant impacts on how the model performs. For example, Wang et al. demonstrated improvement in their models’ performance when data imbalance was addressed by oversampling [45]. ML algorithms may sometimes be very sensitive to missing data, causing an inaccurate output [15,45]. The robustness of ML algorithms to “missingness” typically requires studies on a case-by-case basis, with different strategies to replace or remove observations with missing values tested [15]. Biases that exist when datasets are created may become embedded in the model, leading to the model giving incorrect outputs. A systematic review highlighted that a group of ML models created to detect COVID-19 by analysing chest X-rays and CT scans, which relied on a publicly available dataset of paediatric radiographs as negative comparators, led to models that became efficient at differentiating adult from paediatric images, instead of differentiating COVID-19 patients from those with other causes of pneumonia [48,49]. ML models, like any other device or tool, are not immune to errors or limitations of experimental design, data selection, data processing or model selection and execution. Therefore, several groups have produced recommendations for a reproducible and reliable ML prediction model development in healthcare, including for clinical laboratories [46,50,51,52,53,54].

The International Federation of Clinical Chemistry and Laboratory Medicine (IFCC) has created a working group to address potential pitfalls in the field and, in 2023, the group published recommendations for best practices for the use of ML in laboratory medicine [46]. These recommendations are aimed at improving the quality of published ML models and to ensure that the models are appropriately used, and their outputs are valid, reproducible and reliable. For results to be reproducible, studies should include detailed methodology on the ML model and the specific hardware and software used, as well as the training data sets. The recommendations, however, did not cover how to integrate the ML models into workflows or how the models should be regulated.

There exists a danger that creators view ML as being inherently better than current approaches and seek to incorporate its use even though it may not be accurate. The IFCC working group suggested that laboratories consider whether rules-based approaches or algorithms may be better suited for a given problem before attempting to introduce ML [6].

Additionally, some models function as black-box models, and it is not possible to decipher how the model outputs are determined. If such a model produces erroneous outputs, this lack of explainability could impede detection [55]. A companion concern is of liability for unsatisfactory outcomes because of an erroneous output from a model [56]. Laboratories will have to know the deficiencies in the model and implement measures to mitigate them. Laboratory professionals using the ML model as part of their role in QC or result validation may not have expertise in statistical modelling and thus may not easily identify issues with the model. Two recent studies have highlighted that while healthcare professionals understand the potential of AI to revolutionise the healthcare sector, many fear the potential negative impacts [57,58]. These studies have also shown that healthcare professionals have other ethical concerns, including data security [57,58]. There also exists the difficulty of diverting funding from more pressing needs in the healthcare budget.

## 6. Regulation

The development of AI has been largely unregulated so far. Many countries, including the UK [59] and the EU [60], are now establishing frameworks. The British Medicines and Healthcare products Regulatory Agency (MHRA) policy paper from April 2024 considers the use of AI in healthcare as being within the MHRA’s remit for regulation and will be providing further guidance [61]. The Food and Drug Administration (FDA) of the USA has published a commitment to increase its capacity to understand and regulate the use of AI and ML in the delivery of patient care [62].

## 7. Future

ML models have shown higher accuracy at detecting errors and patterns compared to human validators in studies. ML and AI may free up time for laboratory scientists to work on other aspects of the role and achieve cost, workflow and productivity efficiencies. However, as can be seen from the studies discussed, so far, most ML models have not been tested on real-life, real-time patient data. Therefore, studies demonstrating reproducible real-life, real-time use are required before wider use in laboratory quality control and error detection can be recommended. The development of best practice recommendations and checklists for creating reliable and reproducible ML models is a welcome step, and so are the evolving regulatory aspects.

ML, as well as AI more broadly, is rapidly advancing, with evolving applications. With the increasing availability of higher computing power, sophisticated ML algorithms can be applied to the analysis of more complex data [63,64]. We anticipate increasingly advanced ML models incorporating many different error detection scenarios discussed in this review and perhaps some not even trialled or envisioned yet. Considering the developments so far and the pace of the progress in the field, it would not be surprising to envisage the ML models of the near future contributing to several aspects of a laboratory’s analytical quality, such as the detection of bias, random errors, flagging potential interference and contaminated samples, samples analysed with a delay, wrong-blood-in-tube errors and perhaps a few more not yet thought or trialled aspects (Figure 2). Additionally, we anticipate the incorporation of ML capabilities into commercially available productivity packages, laboratory information management systems and middleware, which may facilitate widespread acceptance and adoption.

## 8. Conclusions

The examples from the literature demonstrate the capability of ML in error detection and QC in medical laboratories. Because the ML studies are addressing different questions and are carried out in different circumstances, using different methods, the direct comparison of models is not possible. We remain optimistic that laboratory professionals will be using such approaches in the future, hopefully with great benefit to the service offered to patients and clinicians. The creation of regulatory frameworks, working groups and best practice recommendations in the field are welcome moves. However, considering that we are in the early days of this rapidly evolving field, and many, if not most, ML models described have only been tested in studies rather than on real-time patient data, due diligence, focusing on safe, standardised, ethical and regulation-compliant use of ML will be essential.

## Figures and Tables

**Figure 1 diagnostics-14-01808-f001:**
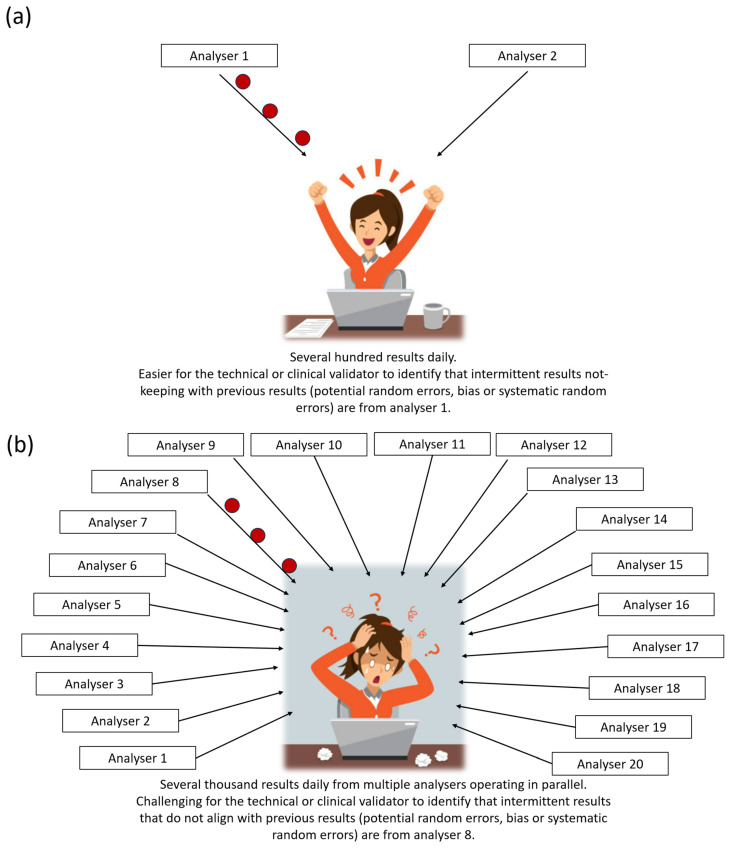
(**a**) Fewer analysers and fewer samples in a smaller laboratory, making pattern recognition and error detection by human technical or clinical validator easier. (**b**) Multiple data streams from numerous analysers and large amounts of data present challenges for the human validator in pattern recognition or error detection.

**Figure 2 diagnostics-14-01808-f002:**
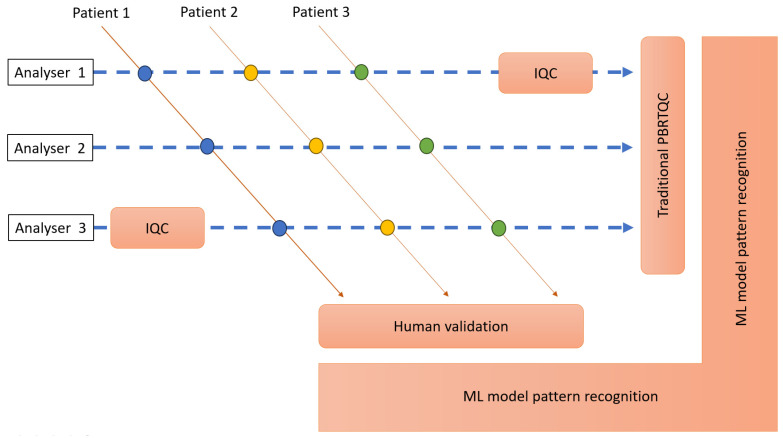
Contribution to analytical quality by traditional IQC methods, result validation by humans, traditional PBRTQC, and potential future ML model-based pattern recognition.

## Data Availability

Not applicable.

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
