# Peer review of "Machine Learning for Patient-Based Real-Time Quality Control (PBRTQC), Analytical and Preanalytical Error Detection in Clinical Laboratory"

_diagnostics, 2024, doi:10.3390/diagnostics14161808_

Round 1

Reviewer 1 Report

Comments and Suggestions for Authors

The authors are reviewing current evidence on the emerging topic of incorporating machine learning into quality control and error detection in medical lavatories. The manuscript is well written and provides a good overview of this topic.

Here are my recommendations:

Page 2: A major limitation of the retrospective nature of standard IQC approaches is the consequence of having to repeat all analyses backwards up to the last correct IQC, including notifying all respective clinicians. Although this is true also for PBRTQC models, the tNAPed surely is far lower. I would recommend, discussing this as well.

Page 5: critical bias” calculated as 0.25 * √((CVi)2+(CVg)2) - if I am not mistaken, this the calculation of the RCV. If so, please mention.

As there are quite a lot of (similar sounding) abbreviations, an index of these abbreviations would be helpful.

I would recommend adding a short conclusion of the PBRTQC approaches just before chapter 3. Where do the models outperform standard iqc, where is iqc still better, which of the presented models was "best", etc.

The same is true for other chapters or subchapters, if more than two studies are discussed, addressing the same issue.

It may be beneficial to add "preanalytical" before "error" in the title.

Page 7 line 296 - typo delts

Author Response

General comment: “The authors are reviewing current evidence on the emerging topic of incorporating machine learning into quality control and error detection in medical lavatories. The manuscript is well written and provides a good overview of this topic.”

Response: “Thank you very much for reviewing the manuscript and providing valuable comments to improve it further.”

Comment 1: "Page 2: A major limitation of the retrospective nature of standard IQC approaches is the consequence of having to repeat all analyses backwards up to the last correct IQC, including notifying all respective clinicians. Although this is true also for PBRTQC models, the tNAPed surely is far lower. I would recommend, discussing this as well."

 Response 1: Thank you for highlighting this. We have expanded our mention of the retrospective nature of traditional IQC processes and potential superiority of PBRTQC for detection of newly introduced bias or imprecision (page 2-3, lines 50-54 and lines 61-62).

Comment 2: "Page 5: critical bias” calculated as 0.25 * √((CVi)2+(CVg)2) - if I am not mistaken, this the calculation of the RCV. If so, please mention."

Response 2: The calculation that Zhou et al used here is not the RCV calculation (which usually is derived from analytical CV and intra-individual CV). What is written instead is the calculation of what is usually termed "desirable bias" which is derived using intraindividual and between individual CV. We have added a short clarification and reference for this on page 5.

Comment 3: "As there are quite a lot of (similar sounding) abbreviations, an index of these abbreviations would be helpful."

Response 3: Thank you for this suggestion. We have created a list of abbreviations at the end of the manuscript.

Comments 4 and 5: "I would recommend adding a short conclusion of the PBRTQC approaches just before chapter 3. Where do the models outperform standard iqc, where is iqc still better, which of the presented models was "best", etc." 

"The same is true for other chapters or subchapters, if more than two studies are discussed, addressing the same issue."

Response 4 and 5: We have added lines 254 to 259 to summarise that ML augmented PBRTQC outperformed standard PBRTQC in the cited studies in chapter 2. As the studies were carried out under different circumstances, direct comparison of ML models is not possible. We have addressed this in lines 526-528. No discussed study compared the use of ML augmented PBRTQC with standard IQC. We have discussed, in lines 44 to 70, the limitations of standard IQC and of standard PBRTQC and hoped to convey with the main body of the article that PBRTQC theoretically offers an improvement over standard IQC except for the issues highlighted in lines 64-64 "PBRTQC is not suitable for every test, for example low throughput, qualitative or semi-quantitative results; and IQC runs will remain important in scenarios such as post-maintenance or post-calibration checks [4, 8]." and that ML augmentation offers even further improvement.

Comment 6: "It may be beneficial to add "preanalytical" before "error" in the title."

Response 6: Thank you for this suggestion. We have amended the title.

Comment 7: "Page 7 line 296 - typo delts".

Response 7: Thank you for spotting this. We have corrected this to "delta".

Thank you again.

Yours sincerely,

Reviewer 2 Report

Comments and Suggestions for Authors

This is a high quality review of a number of different methods for using AI techniques for real time error detection in clinical lab testing. Several different methods for several different use cases are presented, compared, and contrasted, and the background information and caveats in the field are well-addressed. My only comment is to consider using "Analyst" or an alternative word instead of "Analyser" throughout as it currently feels a little clunky, though this is fairly stylistic and could be left as is if preferred.

Author Response

Comment 1: "This is a high quality review of a number of different methods for using AI techniques for real time error detection in clinical lab testing. Several different methods for several different use cases are presented, compared, and contrasted, and the background information and caveats in the field are well-addressed. My only comment is to consider using "Analyst" or an alternative word instead of "Analyser" throughout as it currently feels a little clunky, though this is fairly stylistic and could be left as is if preferred".

Response 1: Thank you very much. We have replaced/ removed some of the occurrences of analyser: line 92 ("machine"), line 197 (removed), line 240 (removed), line 259 (assay).

Thank you again.

Yours sincerely,
